# Pseudo-outcome Imputation with Post-treatment Variables for Individual Treatment Effect Estimation

## Abstract

The estimation of individual treatment effects (ITE) focuses on predicting the outcome changes that result from a change in treatment. A fundamental challenge in observational data is that while we need to infer outcome differences under alternative treatments, we can only observe each individual's outcome under a single treatment. Existing approaches address this limitation either by training with inferred pseudo-outcomes or by creating matched instance pairs. However, these methods overlook post-treatment variables that not only directly influence outcomes but are also affected by exogenous noise. This oversight prevents existing methods from fully capturing outcome variability, resulting in increased variance in counterfactual predictions. In this paper, we introduce PIPCFR, a novel approach that incorporates post-treatment variables through a teacher-student architecture to improve pseudo-outcome imputation. We analyze the challenges inherent in utilizing post-treatment variables and establish a novel theoretical bound for ITE risk that explicitly connects post-treatment variables to ITE estimation accuracy. Unlike existing methods that ignore these variables or impose restrictive assumptions, PIPCFR learns effective representations that preserve informative components while mitigating bias. Empirical evaluations across diverse datasets demonstrate that PIPCFR achieves significantly lower ITE errors compared to existing methods.

## 1 Introduction

Predicting the causal effect of different treatments or interventions is essential in domains such as medicine, finance, and advertising. While traditional approaches depended on randomized controlled trials (RCTs), recent advances focus on leveraging large-scale observational data to estimate treatment effects. For example, marketing strategists can analyze campaign effectiveness – such as variations in monthly ROI under different advertising campaign – to model how promotional strategies influence profitability. However, the fundamental challenge lies in the fact that we observe each individual under only one treatment, with no direct supervision regarding how an individual's outcome would change if the treatment were different.

Existing works address this challenge by imputing pseudo-outcomes for missing treatments in the training data and using them as supervision. These imputation methods include meta-learners (Curth et al., 2021; Nie & Wager, 2021; Künzel et al., 2019), matching methods (Nagalapatti et al., 2024; Schwab et al., 2018; Kallus, 2020), and generative models (Yoon et al., 2018; Louizos et al., 2017; Bica et al., 2020), and their success depends critically on the quality of the inferred pseudo-outcomes. Recently, (Nagalapatti et al., 2024) propose PairNet, a paired instance-based training strategy that avoids relying on potentially noisy pseudo-outcome supervision by instead creating neighbors for each training instance and working directly with observed factual outcomes.

Despite substantial progress in recent years, existing methods face a critical limitation: they overlook post-treatment variables that not only influence outcomes but are also affected by exogenous noise, particularly when the outcome is sensitive to this noise, which leads to increased variance in counterfactual predictions. Consider a real-world advertising example illustrated in Figure 1a. When analyzing the causal effect of an advertising campaign, marketers typically collect user behavior metrics that combine pre-campaign user states (e.g., engagement levels before campaign launch,

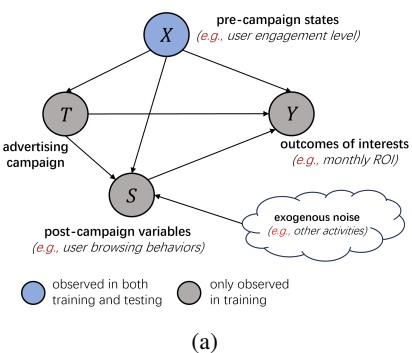

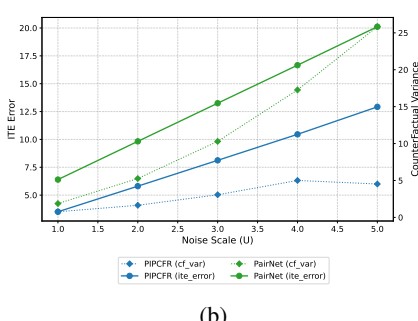

(a)

(b)

Figure 1: (a) A real-world advertising example. (b) The performance of PIPCFR and PairNet on synthetic data: as noise $U$ increases, PairNet shows greater counterfactual variance and ITE errors. In contrast, PIPCFR significantly reduces counterfactual variance, thereby achieving lower ITE errors.

reflecting user preferences that act as confounding variables) and post-campaign variables (e.g., user browsing behaviors after the campaign). These post-campaign variables ($S$) are simultaneously influenced by multiple factors: the pre-campaign variables ($X$), the treatment ($T$), and exogenous noise such as concurrent marketing activities. Together, these variables influence outcomes of interest ($Y$) such as monthly ROI.

Unobserved exogenous noise can indirectly affect outcomes through post-treatment variables, thereby introducing substantial variance into counterfactual predictions. Therefore, without accounting for these post-treatment variables, existing methods cannot fully capture outcome variability, resulting in increased uncertainty in counterfactual estimates. Figure 1b confirms this issue in PairNet, a recent state-of-the-art method: as exogenous noise increases, PairNet shows greater variance in counterfactual predictions, leading to significantly higher ITE estimation errors.

In this paper, we propose PIPCFR, an alternative training strategy that incorporates post-treatment variables to improve pseudo-outcome imputation. Unlike existing methods that either remove post-treatment variables (Zhu et al., 2024) or impose restrictive assumptions like the front-door criterion (Xu et al., 2023), our approach learns effective representations that preserve informative components while mitigating bias. PIPCFR employs a teacher-student architecture where the teacher network uses both pre-treatment covariates ($X$) and post-treatment variables ($S$) as inputs, while the student network uses only $X$. The student learns by using the counterfactual outcomes generated by the teacher as its own counterfactual labels. We analyze the challanges of using post-treatment variables and derive a novel theoretical bound for ITE risk that explicitly connects post-treatment variables to estimation accuracy. As shown in Figure 1b, PIPCFR significantly reduces counterfactual variance, thereby achieving lower ITE estimation errors. Our experiments show that PIPCFR outperforms existing methods across diverse datasets, with notable improvements in challenging scenarios with high exogenous noise and complex temporal dependencies. Furthermore, PIPCFR is compatible with various ITE estimation techniques, consistently improving their performance when combined.

In summary, our contributions are:

1. We introduce PIPCFR, a novel approach that imputes pseudo-outcomes using post-treatment variables to reduce the variance of counterfactual predictions while mitigating potential bias.

2. We establish a theoretical bound for ITE risk that explicitly connects post-treatment variables to estimation accuracy, providing principled guidance for algorithm design.

3. We show that PIPCFR significantly outperforms existing ITE methods across diverse datasets, particularly in challenging scenarios with high exogenous noise and complex temporal dependencies.

## 2 RELATED WORK

**Pseudo-outcomes Imputation** *Meta-Learners* employ a two-stage approach: first training a base model on observational data, then constructing an ITE model using derived pseudo-outcomes.

X-Learner (Künzel et al., 2019) combines predictions from separate treatment groups. DR-Learner (Kennedy, 2023) incorporates propensity scores for doubly robust estimates. R-Learner (Nie & Wager, 2021) directly learns treatment effects using Robinson's decomposition. However, these methods suffer from error propagation between stages and ignore post-treatment variables. *Matching Methods* (Schwab et al., 2018; Kallus, 2020; Iacus et al., 2012) impute outcomes from similar instances using strategies like propensity score matching and covariate-based distance measures. While PairNet (Nagalapatti et al., 2024) improves upon these by using only observed outcomes and creating neighbors, it still depends heavily on accurate distance metrics. Like meta-learners, matching methods also fail to account for post-treatment variables. *Generative Methods* synthesize pseudo-outcomes through various approaches. GANITE (Yoon et al., 2018) employs GANs to generate counterfactuals, SciGAN (Bica et al., 2020) extends this to continuous treatments, while other works explore Gaussian Processes (Zhang et al., 2020) and Variational Autoencoders (Rissanen & Marttinen, 2021).

**Post-treatment Variables in Causal Inference**  In causal inference, existing methods primarily handle post-treatment variables in two ways. The first employs front-door adjustment (Jeong et al., 2022; Wienöbst et al., 2022; Xu et al., 2023), which require post-treatment variables to satisfy the front-door criterion (Pearl, 2009) – for instance, the variables must intercept all directed paths from treatment to outcome. This requirement limits practical application in real-world scenarios where such strict conditions rarely hold. The second approach attempts to disentangle post-treatment variables from observed variables and systematically remove their influence (Zhu et al., 2024; Acharya et al., 2016; Elwert & Winship, 2014; King & Zeng, 2006). While this strategy effectively avoids post-treatment bias by deliberately not controlling for these variables, it inevitably discards valuable information contained within $S$ that could improve estimation accuracy. Unlike these existing works, our method neither rely on the restrictive front-door criterion nor completely discard post-treatment information. Instead, PIPCFR extracts effective representations of post-treatment variables while simultaneously mitigating post-treatment bias.

## 3 PRELIMINARY

We follow the Neyman-Rubin potential outcomes framework (Rubin, 2005), where an individual with observed covariates $x \in \mathcal{X}$, when subjected to a treatment $t \in \mathcal{T}$, exhibits an outcome $Y(t) \in \mathbb{R}$. We denote post-treatment variables as $s \in \mathcal{S}$. We consider binary treatment ($\mathcal{T} = \{0, 1\}$) in this paper. During training, we have an observational dataset $\mathcal{D} = \{x_i, t_i, s_i, y_i\}_{i=1}^{N}$ that includes post-treatment variables, whereas during testing, we must infer ITE based solely on pre-treatment covariates $x$. Samples $(x, t, s, y)$ are drawn from a joint distribution $p(x, t, s, y)$. We denote the covariate distribution as $p(x)$ and the conditional distribution $p(x|t)$ as $p_t(x)$. Define $p_t(x, s) = p(x, s \mid t)$. The marginal treatment distribution is $u_t = p(T = t)$. Let the true outcome function $f_t^*(x) = \mathbb{E}[Y(t) \mid x]$. Our goal is to learn a model $\hat{\tau}(x)$ that estimates the change in outcome $\tau^*(x) = f_1^*(x) - f_0^*(x)$ on a testing sample $x \sim p(x)$. Solving this problem requires us to minimize the following **ITE risk**: $\mathbb{E}_{x \sim P(X)} \left[ (\tau^*(x) - \hat{\tau}(x))^2 \right]$.

**Assumptions.**  Following prior work, we make the following standard assumptions: A1 *Overlap:* Every individual has a non-zero probability of being assigned any treatment, i.e., $p(t|x) \in (0, 1); \forall x, t$. A2 *Stable Unit Treatment Value Assumption:* The outcomes for any sample $i$ are independent of treatment assignments on other samples $j \neq i$. A3 *Unconfoundedness:* The observed covariates block all backdoor paths between treatments and outcomes.

It is worth noting that the assumptions in this paper are the basic assumptions of causal inference; we made no additional assumptions about S or the functional form.

We denote the representation of post-treatment variables as $\psi_\eta : \mathcal{S} \rightarrow \mathcal{R}$. We define *outcome predictor* as $f : \mathcal{X} \times \mathcal{T} \rightarrow \mathbb{R}$ for predicting the outcomes and *pseudo-outcome predictor* as $q : \mathcal{X} \times \mathcal{T} \times \mathcal{R} \rightarrow \mathbb{R}$ for imputing the pseudo-outcomes. To simplify notations, we denote $\phi = \psi_\eta(s)$ as the *post-treatment representation*. We also denote $f_t(x) = f(x, t)$ and $q_t(x, \phi) = q(x, t, \phi)$. We define $p_t(\phi \mid x) = p(\phi \mid x, t)$.

**Definition 1.** *The error residual*

$$r_t(x) = f_t(x) - f_t^*(x), \quad \tilde{r}_t(x, s) = q_t(x, \psi_\eta(s)) - f_t^*(x), \quad \ddot{r}_t(x, s) = f_t(x) - q_t(x, \psi_\eta(s)).$$

$$(1)$$

**Definition 2.** *The factual errors of $f$ are:*

$$\epsilon_F^t = \int_{\mathcal{X}} r_t(x)^2 p_t(x) dx, \quad \epsilon_F = \sum_t u_t \epsilon_F^t, \tag{2}$$

**Definition 3.** *The counterfactual errors of $f$ and $q$ are:*

$$\epsilon_{CF}^t = \int_{\mathcal{X}} r_t(x)^2 p_{1-t}(x) dx, \quad \epsilon_{CF} = \sum_t u_{1-t} \epsilon_{CF}^t,$$

$$\tilde{\epsilon}_{CF}^t = \int_{\mathcal{X}} \tilde{r}_t(x,s)^2 p_{1-t}(s,x) dx, \quad \tilde{\epsilon}_{CF} = \sum_t u_{1-t} \tilde{\epsilon}_{CF}^t. \tag{3}$$

**Definition 4.** *The expected difference of $f$ and $q$ on counterfactual predictions are:*

$$\ddot{\epsilon}_{CF}^t = \int_{\mathcal{X}} \ddot{r}_t(x,s)^2 p_{1-t}(s,x) dx, \quad \ddot{\epsilon}_{CF} = \sum_t u_{1-t} \ddot{\epsilon}_{CF}^t. \tag{4}$$

**Definition 5.** *The ITE risk is defined in terms of residuals as:*

$$\epsilon_{ITE} = \int_{\mathcal{X}} (r_1(x) - r_0(x))^2 p(x) dx$$

**Definition 6.** *The ITE estimation of the outcome predictor is:* $\quad \hat{\tau}(x) := f_1(x) - f_0(x).$

**Proposition 1.** *The ITE risk can be decomposed into a sum of factual error, counterfactual error, and error residuals as follows:*

$$\epsilon_{ITE} = \epsilon_F + \epsilon_{CF} - 2\mathbb{E}_{x,t \sim P(X,T)}[r_t(x) r_{1-t}(x)]. \tag{5}$$

## 4 METHODOLOGY

Existing methods often overlook post-treatment variables. These variables are frequently influenced by external noise and influence outcomes. This oversight leads to an increased variance in counterfactual predictions. To address this, we propose a new approach that leverages the representations of post-treatment variables to impute pseudo-outcomes.

### 4.1 PSEUDO-OUTCOME IMPUTATION WITH POST-TREATMENT VARIABLES (PIP)

In our approach, the pseudo-outcome predictor $q$ takes both pre-treatment covariates and representations of post-treatment variables as inputs and predicts counterfactual outcomes. These predicted outcomes are then used to train the outcome predictor $f$. Using the pseudo-outcomes generated by $q$, we propose an alternative training loss called PIP loss ($\epsilon_{PIP}$) to minimize the ITE risk as follows:

$$\epsilon_{PIP} = \epsilon_F + \ddot{\epsilon}_{CF} - 2\mathbb{E}_{x,t,s \sim p(x,t,s)}[r_t(x) \ddot{r}_{1-t}(x,s)], \tag{6}$$

where $\ddot{\epsilon}_{CF}$ (Definition 4) represents the expected difference between the counterfactual predictions of $q$ and $f$, while $\ddot{r}_{1-t}(x,s)$ (Definition 1) denotes the error residual between the counterfactual predictions of $q$ and $f$.

To optimize the PIP loss (6), we need to address two issues. First, we need to ensure the quality of pseudo-outcomes inferred by $q$, otherwise the imputed values cannot benefit the generalization ability. Second, directly using post-treatment variables as input will introduce post-treatment bias in the estimation. The negative effects of controlling post-treatment variables have been investigated in prior research (Acharya et al., 2016; Elwert & Winship, 2014; Zhu et al., 2024). We present a simple study to illustrate the post-treatment bias in our setting.

**Example 1.** *Consider the structural equation model:*

$$X \leftarrow \mathcal{N}(0, \sigma_x^2), \quad T^* \leftarrow X + \mathcal{N}(0, \sigma_t^2), \quad T \leftarrow \mathbf{1}(T^* > 0),$$

$$u_s \leftarrow \mathcal{N}(0, \sigma_u^2), \quad S \leftarrow X + \alpha_1 T + u_s, \quad Y \leftarrow X + \alpha_2 T + S + \mathcal{N}(0,1), \tag{7}$$

*The parameter $\sigma_x, \sigma_t, \sigma_u$ defines the scale of random variation within the model. The variable $u_s$ is an unobserved exogenous noise that influences $S$. It is drawn from $\mathcal{N}(0, \sigma_u^2)$ and is independent of the treatment $T$.*

*Consider a sample $(x, t, s, y(t))$. To predict the counterfactual outcome $y(1-t)$, we can:*

(a) *regress from* $(X,T)$: $\hat{y}(1-t) - y(1-t) \xrightarrow{d} \mathcal{N}(0, \sigma_u^2 + 1)$.

(b) *regress from* $(X,T,S)$: $\hat{y}(1-t) - y(1-t) \xrightarrow{d} \mathcal{N}((2t-1)\alpha_1, 1)$.

(c) *regress from* $(X,T,u_s)$: $\hat{y}(1-t) - y(1-t) \xrightarrow{d} \mathcal{N}(0, 1)$,

*where $\xrightarrow{d}$ denotes the convergence in distribution.*

Please refer to Appendix section A.4 for the proof. This example demonstrates that: (a) When predicting counterfactual outcomes based solely on pre-treatment covariates $X$ and treatment $T$, the variance of the prediction error is influenced by $\sigma_u$. Higher values of $\sigma_u$ result in greater prediction uncertainty. (b) While incorporating the post-treatment variable $S$ as input can reduce this variance, it introduces bias into the predictions. This bias occurs because $S$ is influenced by the treatment $T$, causing its distribution to vary between treatment and control groups. (c) In contrast, using the variable $u_s$ as a feature offers the best of both worlds: it reduces the variance of counterfactual predictions without introducing bias, since the distribution of $u_s$ remains invariant across different treatments. Although $u_s$ is typically unobservable, we can potentially extract information about $u_s$ from the observable $S$.

Therefore, to construct effective pseudo labels for the PIP loss (6), we need to learn representations of post-treatment variables that extract useful variance-reducing information while eliminating components that would introduce bias.

To achieve this, we investigate the gap between the ITE risk $\epsilon_{ITE}$ and our PIP loss $\epsilon_{PIP}$, and provide a novel bound for the ITE risk that explicitly connect post-treatment variables to estimation accuracy.

**Theorem 1.** *Assume that exists $\delta > 0$ such that $\mathbb{E}_{x,t,s \sim p(x,t,s)}[r_t(x)\ddot{r}_{1-t}(x,s)] \leq \delta\tilde{\epsilon}_{CF}$, we have*

$$\epsilon_{ITE} \leq \underbrace{\epsilon_{PIP}}_{\text{PIP Loss}} + \underbrace{\mathbb{E}_{x \sim p(x)}[B \cdot IPM_G(p_0(\phi \mid x), p_1(\phi \mid x))]}_{\text{Post-Treatment Bias}} + \underbrace{(2\delta+1)\tilde{\epsilon}_{CF}}_{\text{Generalization Error of } q} + 2\sum_{t=0,1} u_t \left[\sqrt{\epsilon_F^t Q_t(\psi_\eta, q)}\right],$$

(8)

*where $Q_t(\psi_\eta, q) = \int_{\mathcal{X}} \int_{\mathcal{S}} [f^*(x) - q(x, 1-t, \psi_\eta(s))]^2 p_t(s,x) dx$ can be regarded as evaluating how well predictor $q$ models (or represents) counterfactual outcome information, where this modeling is facilitated by the introduction of $s$. Additionally, there exists a constant $B$ such that $\frac{1}{B}\bar{r}_t(x,\phi) \in G$ where $\bar{r}_t(x,\phi) = f_t(x) - q_t(x,\phi)$.*

Please refer to Appendix section A.2 for the proof. Theorem 1 shows that the ITE risk is upper bounded by the sum of: (1) the PIP Loss, (2) an Integral Probability Metric (IPM) that measures the distributional distance of representations $\phi$ across treatment groups—a source of post-treatment bias that our model aims to minimize to encourage $\phi \perp\!\!\!\perp t \mid x$, and (3) the generalization error of the pseudo-outcome predictor $q$. The last term measures the additional information about counterfactual outcomes gained by the pseudo-outcome predictor $q$ after incorporating post-treatment variables; when $q$ is optimally trained, this term depends only on the inherent characteristics of the data.

## 4.2 PRACTICAL ALGORITHM

Building upon our theoretical results, we propose PIPCFR, an end-to-end algorithm for ITE estimation. Motivated by Theorem 1, we minimize the upper bound in (8) in order to minimize the ITE risk.

**Minimizing Post-treatment Bias** First, to eliminate the post-treatment bias, we need to minimize the IPM term in (8). However, in the observational data, we only have representations $\phi$ under one treatment, and cannot access both $p_0(\phi \mid x)$ and $p_1(\phi \mid x)$ at the same time. A straightforward method to compute this IPM term is to use matching strategy by creating neighbors to estimate the representations $\phi$ under the missing treatment, but this method is time-consuming and highly relies on distance metrics. We propose an alternative solution that instead minimizes the Kullback–Leibler (KL) divergence $KL(p(t \mid x)\|p(t \mid x, \phi))$.

**Proposition 2** (Relation between IPM and KL Divergence). *Let $G$ be the family of norm-1 functions in a Reproducing Kernel Hilbert Space (RKHS). Assume $G$ is generated by a normalized kernel. The*

*Integral Probability Metric (IPM) between two distributions $p_0(\phi \mid x)$ and $p_1(\phi \mid x)$) is bounded by their conditional KL divergence as follows:*

$$IPM_G(p_0(\phi \mid x), p_1(\phi \mid x)) \leq \sqrt{\frac{2}{\pi_0 \pi_1} \mathbb{E}_{\phi \sim p(\phi|x)} \big[ KL(p(t \mid x, \phi) \| p(t \mid x)) \big]},$$

*where $\pi_0 = p(t = 0 \mid x), \pi_1 = p(t = 1 \mid x)$. Specifically, if $\mathbb{E}_{s,x \sim p(s,x)}[KL(p(t \mid x, \phi) \| p(t \mid x))] = 0$, then $\mathbb{E}_{x \sim p(x)}[IPM_G(p_0(\phi \mid x), p_1(\phi \mid x))] = 0$.*

Please refer to Appendix section A.3 for the proof. Minimizing the KL term indicates maximizing the conditional independence between $\phi$ and $t$ given $x$. To achieve this, we introduce two propensity score models: $g(t, x) = \hat{p}(t \mid x)$ and $\tilde{g}(t, x, \phi) = \hat{p}(t \mid x, \phi)$, which are trained to predict treatments using the following objective:

$$\min_{g, \tilde{g}} \mathcal{L}_p = -\frac{1}{N} \sum_{i=1}^{N} \left[ \log g(t_i, x_i, \psi_\eta(s_i)) + \log \tilde{g}(t_i, x_i) \right]. \tag{9}$$

We then optimize the parameter of post-treatment representation $\eta$ to minimize the KL loss:

$$\min_{\psi_\eta} \mathcal{L}_{KL} = \frac{1}{N} \sum_{i=1}^{N} \left[ \gamma \cdot \sum_t g(t, x_i)(\log \tilde{g}(t, x_i, \psi_\eta(s_i)) - \log g(t, x_i)) \right], \tag{10}$$

where $\gamma$ is a hyper parameter. This induces conditional independence by ensuring that a propensity score model does not gain any additional information about treatment when given access to $\psi_\eta(s_i)$.

**Minimizing Generalization Error of** $q$ Second, to ensure the quality of the pseudo-outcomes, it's necessary to minimize the generalization error of $q$. In principle, we can employ any existing causal inference algorithms for this purpose. As shown in Figures 2 and 3, current methods typically rely on a matching approach. However, this approach has a critical dependency on appropriate distance metrics, which can lead to biased sample selection if unsuitable. Moreover, the matching process becomes computationally prohibitive when applied to large-scale datasets due to the substantial time required to identify suitable counterfactual samples. To address these issues, we use a deep learning approach that is capable of predicting counterfactual outcomes. Specifically, we propose using a TARNet-like method to construct the pseudo-outcome predictor.

In this paper, we adopt CFRNet (Shalit et al., 2017) to minimize $\tilde{\epsilon}_{CF}$. The learning objective of CFRNet combines a weighted factual loss with an IPM distance that measures the divergence of covariate representations between treatment and control groups. We define the pseudo-outcome predictor as $q(x, t, \phi) = h(\psi_\alpha(x), t, \phi)$, where $\psi_\alpha$ is a representation extraction function shared across treatments, and $h$ is the treatment-specific outcome prediction function. The learning objective is as follows:

$$\min_{h, \psi_\alpha, \psi_\eta} \mathcal{L}_y = \frac{1}{N} \sum_{i=1}^{N} \beta_i \cdot (h(\psi_\alpha(x_i), t_i, \psi_\eta(s_i)) - y_i)^2 + IPM_G\left(\{\psi_\alpha(x_i)\}_{i:t_i=0}, \{\psi_\alpha(x_i)\}_{i:t_i=1}\right),$$

$$\text{with} \quad \beta_i = \frac{t_i}{2u} + \frac{1 - t_i}{2(1 - u)}, \quad \text{where} \quad u = \frac{1}{N} \sum_{i=1}^{N} t_i. \tag{11}$$

Through optimization of the generalization error term, representations $\phi$ extract information from post-treatment variables that provides additional predictive power for outcomes conditional on $x$, effectively capturing the influence of unobserved exogenous noise $U$.

**Minimizing PIP Loss** Third, we minimize the $\epsilon_{PIP}$ term. We use a TARNet to construct the outcome predictor $f$. The empirical loss can be written as:

$$\min_f \mathcal{L}_{pip} = \frac{1}{N} \sum_{i=1}^{N} (f(x_i, t_i) - y_i)^2 + (f(x_i, 1 - t_i) - q(x_i, 1 - t_i, \psi_\eta(s_i)))^2 \tag{12}$$
$$- 2 (f(x_i, t_i) - y_i) (f(x_i, 1 - t_i) - q(x_i, 1 - t_i, \psi_\eta(s_i))).$$

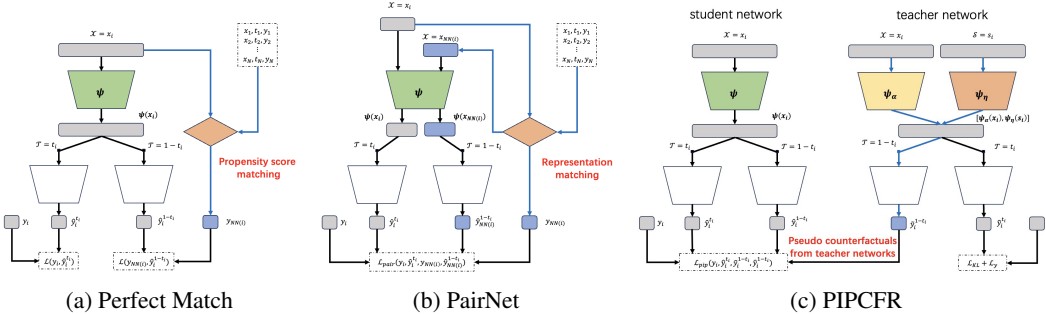

|  |  |  |
| :---: | :---: | :---: |
| (a) Perfect Match | (b) PairNet | (c) PIPCFR |

Figure 2: Comparison of architectures. Perfect Match augments samples within a minibatch with their propensity matched nearest neighbours. PairNet minimizes losses over observed instance pairs that are close in the covariate space while having different treatments. PIPCFR imputes pseudo counterfactuals by incorporating post-treatment variables through a teacher-student architecture.

The PIPCFR architecture is shown in Figure 2c. Since the pseudo-outcome predictor $q$ provides pseudo-labels for the outcome predictor $f$, we refer to $q$ as the **teacher network** and $f$ as the **student network**. We train our teacher-student architecture models by jointly optimizing the objectives (9), (10), (11) and (12) using stochastic gradient descent. Please refer to Appendix C for more details.

## 5 EXPERIMENTS

In this section, we aim to answer the following research questions:

- RQ1: How does PIPCFR perform compared to state-of-the-art methods in ITE estimation?

- RQ2: How robust is PIPCFR when post-treatment variables exhibit complexity and noise?

- RQ3: How does PIPCFR perform when combined with other methods?

- RQ4: How sensitive is PIPCFR to the choice of hyper-parameters?

**Dataset.** We evaluate our approach on two benchmark datasets (IHDP (Hill, 2011) and News (Johansson et al., 2016)) and a synthetic dataset. For IHDP and News, post-treatment variables are generated following (Cheng et al., 2021). For the synthetic dataset, we simulate a temporal causal system with interacting variables over time. Additional details are provided in Appendix B.

**Baseline.** We compare PIPCFR with several competitive baselines, as shown in Table 1. (1) *Tree Learners*: causal forest (ATHEY et al., 2019), which builds upon traditional random forests to estimate ITE. (2) *Meta-learners* such as XLearner (Künzel et al., 2019) that learn ITE directly after imputing pseudo-outcomes for missing treatments. (3) *Representation-learning methods* like TARNet (Shalit et al., 2017), CFRNet (Shalit et al., 2017), and DRCFR (Hassanpour et al., 2019). These approaches share representations between different treatment heads while learning treatment-specific estimators with varied regularization techniques. (4) *Propensity-score Learner* such as DragonNet (Shi et al., 2019) is a doubly robust method that imposes weighted factual losses. (5) *Matching methods* including PairNet (Nagalapatti et al., 2024) and PerfectMatch (Schwab et al., 2018), which employ matching strategies by creating neighboring samples to impute outcomes for missing treatments. Additionally, we consider two extended baselines: PairNet+S and PerfectMatch+S, which incorporate post-treatment variables $S$ when matching neighboring samples.

**Metrics.** We evaluate ITE risk on a dataset $D_{tst}$ using the Presicion in Estimating Heterogeneous Effects ($\epsilon_{PEHE}$) (Johansson et al., 2016) defined as: $\sqrt{\frac{1}{|\mathcal{D}_{\text{test}}|}\sum_{i=1}^{|\mathcal{D}_{\text{test}}|}(\hat{\tau}(x_i) - \tau^*(x_i))^2}$. We quantify PEHE (in) error for training instances and PEHE (out) error for testing instances. To ensure the reliability of the results, we sampled 50 datasets ($\mathcal{D}_{\text{train}}, \mathcal{D}_{\text{test}}$) from the data distribution $p(x, t, s, y)$ for training and testing, and reported the mean and standard deviation.

Table 1: RQ1: The performance of PIPCFR compared to baselines evaluated using PEHE error. The table shows mean values and standard deviation across 50 runs. Overall, PIPCFR demonstrates superior performance among all methods.

| | | NEWS | | IHDP | | Synthetic | |
|---|---|---|---|---|---|---|---|
| | | $\epsilon_{\text{PEHE in}}$ | $\epsilon_{\text{PEHE out}}$ | $\epsilon_{\text{PEHE in}}$ | $\epsilon_{\text{PEHE out}}$ | $\epsilon_{\text{PEHE in}}$ | $\epsilon_{\text{PEHE out}}$ |
| Tree Learners | Causal Forest (ATHEY et al., 2019) | 1.39 ± 0.00 | 1.40 ± 0.00 | 2.90 ± 0.01 | 2.92 ± 0.02 | 4.67 ± 0.00 | 4.73 ± 0.01 |
| Meta Learners | XLearner (Künzel et al., 2019) | 1.09 ± 0.00 | 1.12 ± 0.00 | 3.64 ± 0.06 | 3.90 ± 0.07 | 3.97 ± 0.29 | 4.09 ± 0.29 |
| | RLearner (Nie & Wager, 2021) | 1.33 ± 0.02 | 1.27 ± 0.02 | 8.85 ± 0.08 | 6.77 ± 0.10 | 19.74 ± 1.68 | 19.49 ± 1.69 |
| | DRLearner (Kennedy, 2023) | 4.51 ± 0.08 | 3.82 ± 0.12 | 5.04 ± 0.23 | 4.99 ± 0.27 | 6.78 ± 1.04 | 6.93 ± 1.07 |
| Rep. Learners | DRCFR (Hassanpour et al., 2019) | 0.84 ± 0.01 | 0.92 ± 0.01 | 2.98 ± 0.10 | 3.00 ± 0.11 | 7.80 ± 0.02 | 7.97 ± 0.02 |
| | ESCFR (Wang et al., 2024) | 1.78 ± 0.01 | 1.78 ± 0.01 | 2.93 ± 0.01 | 2.95 ± 0.01 | 3.93 ± 0.20 | 4.09 ± 0.20 |
| | TARNet (Shalit et al., 2017) | 0.69 ± 0.01 | 0.76 ± 0.01 | 4.40 ± 0.10 | 4.32 ± 0.11 | 6.67 ± 0.14 | 6.60 ± 0.13 |
| | CFRNet(MMD) (Shalit et al., 2017) | 0.84 ± 0.01 | 0.80 ± 0.01 | 3.38 ± 0.10 | 3.37 ± 0.09 | 6.42 ± 0.01 | 6.56 ± 0.01 |
| | CFRNet(WASS) (Shalit et al., 2017) | 0.81 ± 0.01 | 0.85 ± 0.01 | 2.89 ± 0.05 | 2.90 ± 0.04 | 4.96 ± 0.03 | 5.03 ± 0.03 |
| PS. Learner | DragonNet (Shi et al., 2019) | 0.49 ± 0.01 | 0.61 ± 0.01 | 3.68 ± 0.11 | 3.61 ± 0.10 | 5.04 ± 0.03 | 5.10 ± 0.03 |
| Matching | PairNet (Nagalapatti et al., 2024) | 0.95 ± 0.01 | 1.11 ± 0.01 | 4.39 ± 0.01 | 4.43 ± 0.03 | 4.20 ± 0.01 | 4.29 ± 0.01 |
| | PairNet+S | 0.98 ± 0.01 | 1.13 ± 0.01 | 4.43 ± 0.01 | 4.46 ± 0.03 | 3.52 ± 0.01 | 3.62 ± 0.01 |
| | PerfectMatch (Schwab et al., 2018) | 0.96 ± 0.01 | 1.05 ± 0.01 | 13.23 ± 0.05 | 13.22 ± 0.08 | 4.86 ± 0.09 | 5.04 ± 0.09 |
| | PerfectMatch+S | 0.94 ± 0.01 | 1.06 ± 0.01 | 13.21 ± 0.05 | 13.23 ± 0.08 | 6.68 ± 0.17 | 6.86 ± 0.17 |
| | **PIPCFR (MMD)** | 0.30 ± 0.00 | 0.46 ± 0.00 | **1.96 ± 0.01** | **2.00 ± 0.01** | 3.09 ± 0.06 | 3.26 ± 0.06 |
| | **PIPCFR (WASS)** | **0.25 ± 0.00** | **0.44 ± 0.00** | 2.27 ± 0.01 | 2.35 ± 0.02 | **2.88 ± 0.05** | **3.06 ± 0.05** |

## 5.1 RQ1: PIPCFR VS. BASELINES

We present the results in Table 1. PIPCFR consistently outperforms existing methods from five categories of prior techniques for ITE. Meta Learners show poor performance due to their two-staged regression approach. Missing outcomes imputed during the first stage create errors that propagate to the second stage, resulting in suboptimal ITE estimation. Through joint training approaches, representation learners and PS-learners outperform meta-learners by enabling effective information sharing across different treatments. However, a key limitation of these methods is that they lack specific mechanisms to address the variance that emerges from post-treatment variables. Matching methods perform poorly as they depend on distance metrics to create neighboring samples for missing treatments, which can be unreliable when post-treatment variables introduce variance. Notably, even when post-treatment variables ($S$) are incorporated in the matching process (PairNet+S, PerfectMatch+S), these methods still underperform. This confirms our analysis in Section 4.1 that directly using $S$ as input will introduce post-treatment bias. In contrast to these baselines, PIPCFR demonstrates superior performance by effectively extracting useful information from post-treatment variables while simultaneously mitigating post-treatment bias and imputing accurate pesudo-outcomes for missing treatments through the teacher-student architecture.

## 5.2 RQ2: INFLUENCE OF POST-TREATMENT VARIABLES

**Impact of Exogenous Noise**   To evaluate the impact of exogenous noise, we vary the noise scales $\epsilon_u = [1, 2, 3, 4, 5]$ as defined in Appendix B. As shown in Figure 3a, demonstrate a clear pattern: as noise increases, all methods exhibit increasing PEHE error. Notably, PIPCFR consistently outperforms all baselines across different noise scales, demonstrating its robust performance in the presence of significant exogenous noise. More importantly, PIPCFR's performance advantage grows with increasing noise, indicating its superior capability in handling the uncertainty in post-treatment variables compared to existing approaches.

**Impact of Temporally Accumulated Noises**   In real-world scenarios, post-treatment variables can manifest as sequences where noise accumulates over time, making ITE estimation increasingly challenging as the time horizon extends. We consider post-treatment variables as a $K$-step sequence, as defined in the Appendix B. We experiment with a fixed noise scale while varying $K \in [1, 100]$. The results are presented in Figure 3b. As expected, ITE estimation errors increase for all methods as $K$ increases. Notably, PIPCFR achieves lower PEHE error compared to the baselines. Furthermore, PIPCFR's performance advantage actually widens as $K$ increases, demonstrating its superior ability to capture complex temporal dependencies between post-treatment variables and outcomes.

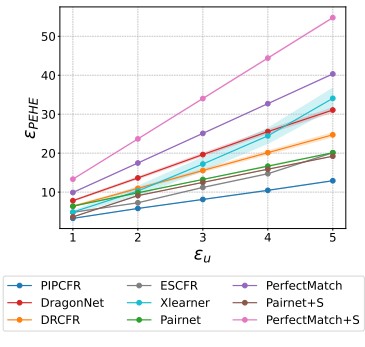

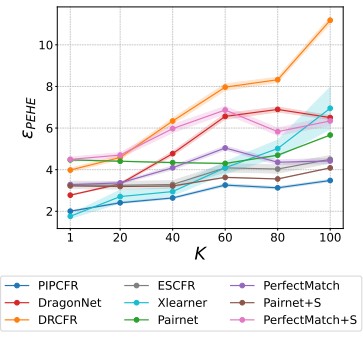

(a) Varying noise scales

(b) Varying timestep

Figure 3: Performance comparison with varying post-treatment variables.

Table 2: RQ 3: Compatibility with existing methods. We combine several ITE methods with our approach and observe that our approach provides a significant performance improvement on $\epsilon_{\text{PEHE out}}$.

|  | DRCFR | | DragonNet | | ESCFR | |
|---|---|---|---|---|---|---|
|  | Baseline | +PIPCFR | Baseline | +PIPCFR | Baseline | +PIPCFR |
| NEWS | $0.92 \pm 0.01$ | $0.44 \pm 0.00$ | $0.61 \pm 0.01$ | $0.40 \pm 0.00$ | $1.78 \pm 0.01$ | $0.41 \pm 0.00$ |
| IHDP | $3.00 \pm 0.11$ | $2.31 \pm 0.01$ | $3.61 \pm 0.10$ | $2.35 \pm 0.02$ | $2.95 \pm 0.01$ | $2.35 \pm 0.02$ |
| Synthetic | $7.80 \pm 0.02$ | $3.31 \pm 0.02$ | $5.04 \pm 0.03$ | $2.95 \pm 0.03$ | $4.09 \pm 0.20$ | $2.80 \pm 0.05$ |

### 5.3 RQ3: COMPATIBILITY WITH EXISTING METHODS

In Section 4.2, we employ CFRNet (Shalit et al., 2017) to minimize the generalization error of the teacher network. However, our approach is fundamentally flexible – in principle, we can utilize any existing causal inference methods to minimize this generalization error, making PIPCFR inherently compatible with a wide range of existing ITE estimation techniques. To demonstrate this compatibility, we conduct experiments in which we replace the CFRNet method with several alternatives, including DRCFR, DragonNet, and ESCFR. The results, presented in Table 2, clearly show that PIPCFR consistently enhances performance when integrated with these existing ITE estimation methods.

### 5.4 RQ4: SENSITIVITY ANALYSIS

We examine the impact of KL loss weight $\gamma$ in (10), as shown in Figure 4. The performance drop at $\gamma = 0$ highlights the importance of conditional independence when learning representations of post-treatment variables. Large $\gamma$ values lead to performance degradation, likely because the KL divergence term dominates the optimization process. The model achieves optimal performance within a stable range of $\gamma \in [0.1, 1]$, showing that PIPCFR is not very sensitive to hyperparameter choice. This stability reduces the need for precise hyperparameter tuning, enhancing the practical applicability of our approach.

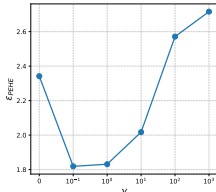

Figure 4: Varying $\gamma$.

## 6 CONCLUSION

In this paper, we addressed a critical limitation in existing ITE estimation methods: the oversight in accounting for post-treatment variables that introduce significant variance in counterfactual predictions. We introduced PIPCFR, a novel approach that leverages post-treatment variables to impute more accurate pseudo-outcomes through a teacher-student architecture. Our theoretical analysis established a new bound for ITE risk that explicitly connects post-treatment variables to estimation accuracy. Experiments demonstrated that PIPCFR consistently outperforms state-of-the-art methods from various categories. We showed that PIPCFR is compatible with existing ITE methods.

Our work opens new directions for causal inference research by highlighting the importance of properly handling post-treatment variables. While promising, this work has limitations, particularly its reliance on standard causal inference assumptions such as unconfoundedness. In the future, several interesting research directions lie ahead. (1) Incorporate PIPCFR into more sophisticated models that account for hidden confounders. (2) Exploring advanced network architectures to enhance the representation learning of post-treatment variables.

## ETHICS STATEMENT

**General Statement:** This research aims to improve methods for causal inference. We believe that causal inference, as a tool for understanding complex systems, is primarily applied in fields such as scientific discovery, medical analysis, and policy making, and is therefore **unlikely to produce direct negative societal impacts**. Nevertheless, we have carefully assessed potential ethical issues.

**Data Use and Privacy:** The benchmark datasets used in this research, including IHDP (Hill, 2011) and News (Johansson et al., 2016), are open-source. The synthetic dataset was generated from a temporal causal model, and its generation procedure is implemented in our released code. All datasets used in our experiments are reproducible by running the scripts provided in the code package. We did not use or share any Personally Identifiable Information (PII) at any point.

**Social Impact and Responsibility:** Researchers have a responsibility to use their skills to benefit society, its members, and the natural environment. Our research aims to minimize potential negative consequences, including threats to health, safety, and privacy. We are committed to ensuring our research outcomes will respect diversity, be used in a socially responsible manner, meet societal needs, and be broadly accessible.

## REPRODUCIBILITY STATEMENT

**Code and hyperparameters:** We provide the code for model training, evaluation, and dataset generation in the supplementary material, together with a comprehensive README. We also discuss implementation details in Appendix section C, including hyperparameter settings, network architecture and operating environments.

**Datasets:** The benchmark datasets used in our experiments (IHDP (Hill, 2011) and News (Johansson et al., 2016)) are open-source. The synthetic dataset was generated from a temporal causal model; the generation procedure is implemented in the released code. Further dataset descriptions and preprocessing details are given in Appendix B. All datasets used in our experiments can be reproduced by running the scripts provided in the code package.

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

# A    PROOFS

## A.1    DEFINITION

**Definition 7.** *The error residual*

$$r_t(x) = f_t(x) - f_t^*(x), \quad \tilde{r}_t(x,s) = q_t(x, \psi_\eta(s)) - f_t^*(x),$$
$$\ddot{r}_t(x,s) = f_t(x) - q_t(x, \psi_\eta(s)), \quad \overline{r}_t(x,\phi) = f_t(x) - q_t(x,\phi). \tag{13}$$

**Definition 8.** *The counterfactual errors are:*

$$\epsilon_{CF}^t = \int_x r_t(x)^2 p_{1-t}(x)dx, \quad \epsilon_{CF} = \sum_t u_{1-t}\epsilon_{CF}^t,$$

$$\overline{\epsilon}_{CF}^t = \int_x \overline{r}_t(x,s)^2 p_{1-t}(s,x)dx, \quad \overline{\epsilon}_{CF} = \sum_t u_{1-t}\overline{\epsilon}_{CF}^t,$$

$$\tilde{\epsilon}_{CF}^t = \int_x \tilde{r}_t(x,s)^2 p_{1-t}(s,x)dx, \quad \tilde{\epsilon}_{CF} = \sum_t u_{1-t}\tilde{\epsilon}_{CF}^t, \tag{14}$$

$$\ddot{\epsilon}_{CF}^t = \int_x \ddot{r}_t(x,s)^2 p_{1-t}(s,x)dx, \quad \ddot{\epsilon}_{CF} = \sum_t u_{1-t}\ddot{\epsilon}_{CF}^t.$$

## A.2    PROOF OF THEOREM 1

**Theorem.** *Assume that exists $\delta > 0$ such that $\mathbb{E}_{x,t,s\sim p(X,T,S)}[r_t(x)\ddot{r}_{1-t}(x,s)] \leq \delta\tilde{\epsilon}_{CF}$, we have*

$$\epsilon_{ITE} \leq \epsilon_{PIP} + \mathbb{E}_{x\sim p(X)}[B \cdot IPM_G(p_0(\phi \mid x), p_1(\phi \mid x))] + (2\delta + 1)\tilde{\epsilon}_{CF} + 2\sum_{t=0,1} u_t \left[\sqrt{\epsilon_F^t Q_t(\psi_\eta, q)}\right], \tag{15}$$

*where $Q_t(\psi_\eta, q) = \int_{\mathcal{X}} \int_{\mathcal{S}} [f^*(x) - q(x, 1-t, \psi_\eta(s))]^2 p_t(s,x)dx$ can be regarded as evaluating how well predictor $q$ models (or represents) counterfactual outcome information, where this modeling is facilitated by the introduction of $s$. Additionally, there exists a constant $B$ such that $\frac{1}{B}\overline{r}_t(x,\phi) \in G$ where $\overline{r}_t(x,\phi) = f_t(x) - q_t(x,\phi)$.*

*Proof.* The decompositin of ITE risk is:

$$\epsilon_{ITE} = \sum_t u_t \int_{\mathcal{X}} (r_1(x) - r_0(x))^2 dp_t(x)$$

$$= u_0 \int_{\mathcal{X}} [r_1(x)^2 + r_0(x)^2 - 2r_1(x)r_0(x)]p_0(x)dx \tag{16}$$

$$+ u_1 \int_{\mathcal{X}} [r_1(x)^2 + r_0(x)^2 - 2r_1(x)r_0(x)]p_1(x)dx.$$

The decompositin of our PIP loss is:

$$\epsilon_{PIP} = \sum_t u_t \int_{\mathcal{X}} \int_{\mathcal{S}} [r_t(x) - \ddot{r}_{1-t}(x,s)]^2 dp_t(s \mid x)dp_t(x)$$

$$= u_0 \int_{\mathcal{X}} r_0(x)^2 ds + u_0 \int_{\mathcal{X}} \int_{\mathcal{S}} \ddot{r}_1(x,s)^2 p_0(s \mid x)ds - 2u_0 \int_{\mathcal{X}} \int_{\mathcal{S}} r_0(x)\ddot{r}_1(x,s)p_0(s \mid x)ds p_0(x)dx$$

$$+ u_1 \int_{\mathcal{X}} r_1(x)^2 + u_1 \int_{\mathcal{X}} \int_{\mathcal{S}} \ddot{r}_0(x,s)^2 p_1(s \mid x)ds - 2u_1 \int_{\mathcal{X}} \int_{\mathcal{S}} r_1(x)\ddot{r}_0(x,s)p_1(s \mid x)ds p_1(x)dx.$$
$$\tag{17}$$

The difference between the ITE risk and our PIP loss is:

$$\epsilon_{ITE} - \epsilon_{PIP}$$

$$= u_0 \int_{\mathcal{X}} [r_1(x)^2 + r_0(x)^2 - 2r_1(x)r_0(x)]p_0(x)dx$$

$$+ u_1 \int_{\mathcal{X}} [r_1(x)^2 + r_0(x)^2 - 2r_1(x)r_0(x)]p_1(x)dx$$

$$- \left\{ u_0 \int_{\mathcal{X}} r_0(x)^2 ds + u_0 \int_{\mathcal{X}} \int_{\mathcal{S}} \ddot{r}_1(x,s)^2 p_0(s \mid x)ds - 2u_0 \int_{\mathcal{X}} \int_{\mathcal{S}} r_0(x)\ddot{r}_1(x,s)p_0(s \mid x)ds p_0(x)dx \right.$$

$$\left. + u_1 \int_{\mathcal{X}} r_1(x)^2 + u_1 \int_{\mathcal{X}} \int_{\mathcal{S}} \ddot{r}_0(x,s)^2 p_1(s \mid x)ds - 2u_1 \int_{\mathcal{X}} \int_{\mathcal{S}} r_1(x)\ddot{r}_0(x,s)p_1(s \mid x)ds p_1(x)dx \right\}$$

$$= u_0 \underbrace{\int_{\mathcal{X}} [r_1(x)^2 - \int_{\mathcal{S}} \ddot{r}_1(x,s)^2 p_0(s \mid x)ds]p_0(x)dx}_{H_1}$$

$$+ u_1 \underbrace{\int_{\mathcal{X}} [r_0(x)^2 - \int_{\mathcal{S}} \ddot{r}_0(x,s)^2 p_1(s \mid x)ds]p_1(x)dx}_{H_2}$$

$$+ 2u_0 \underbrace{\int_{\mathcal{X}} [\int_{\mathcal{S}} r_0(x)\ddot{r}_1(x,s)p_0(s \mid x)ds - r_1(x)r_0(x)]p_0(x)dx}_{H_3}$$

$$+ 2u_1 \underbrace{\int_{\mathcal{X}} [\int_{\mathcal{S}} r_1(x)\ddot{r}_0(x,s)p_1(s \mid x)ds - r_1(x)r_0(x)]p_1(x)dx}_{H_4}$$
$$\tag{18}$$

In the following, we derive $H_1$, $H_2$, $H_3$ and $H_4$ respectively.

$$H_1 = u_0 \int_{\mathcal{X}} \int_{\mathcal{S}} [\int_{\mathcal{S}} (\ddot{r}_1(x,s) + \tilde{r}_1(x,s))^2 p_1(s \mid x) ds - \int_{\mathcal{S}} \ddot{r}_1(x,s)^2 p_0(s \mid x) ds] p_0(x) dx$$

$$= u_0 \int_{\mathcal{X}} \int_{\mathcal{S}} [\int_{\mathcal{S}} [\ddot{r}_1(x,s)^2 + \tilde{r}_1(x,s)^2 + 2\ddot{r}_1(x,s)\tilde{r}_1(x,s)] p_1(s \mid x) ds - \int_{\mathcal{S}} \ddot{r}_1(x,s)^2 p_0(s \mid x) ds] p_0(x) dx$$

$$= u_0 \int_{\mathcal{X}} [\int_{\mathcal{S}} \ddot{r}_1(x,s)^2 (p_1(s \mid x) - p_0(s \mid x)) ds + \int_{\mathcal{S}} \tilde{r}_1(x,s)^2 p_1(s \mid x) ds$$

$$+ 2 \int_{\mathcal{S}} \ddot{r}_1(x,s)\tilde{r}_1(x,s) p_1(s \mid x) ds] p_0(x) dx$$

$$\leq u_0 \int_{\mathcal{X}} [\int_{\mathcal{S}} \ddot{r}_1(x,s)^2 (p_1(s \mid x) - p_0(s \mid x)) ds + (2\delta + 1) \int_{\mathcal{S}} \tilde{r}_1(x,s)^2 p_1(s \mid x) ds] p_0(x) dx$$

$$= u_0 \int_{\mathcal{X}} [\int_{\Phi} \overline{r}_1(x,\phi)^2 (p_1(\phi \mid x) - p_0(\phi \mid x)) ds + (2\delta + 1) \int_{\mathcal{S}} \tilde{r}_1(x,s)^2 p_1(s \mid x) ds] p_0(x) dx \tag{19}$$

$$H_2 \leq u_1 \int_{\mathcal{X}} [\int_{\mathcal{S}} \overline{r}_0(x,\phi)^2 (p_0(\phi \mid x) - p_1(\phi \mid x)) ds + (2\delta + 1) \int_{\mathcal{S}} \tilde{r}_0(x,s)^2 p_0(s \mid x) ds] p_1(x) dx$$

$$H_1 + H_2 \leq \mathbb{E}_x [B \cdot IPM_G(p_0(\phi \mid x), p_1(\phi \mid x))] + (2\delta + 1)\tilde{\epsilon}_{CF} \tag{20}$$

$$H_3 = 2u_0 \int_{\mathcal{X}} [\int_{\mathcal{S}} r_0(x)\ddot{r}_1(x,s) p_0(s \mid x) ds - r_1(x)r_0(x)] p_0(x) dx$$

$$= 2u_0 \int_{\mathcal{X}} [\int_{\mathcal{S}} r_0(x)\ddot{r}_1(x,s) p_0(s \mid x) ds - \int_{s} r_1(x)r_0(x) p_0(s \mid x) ds] p_0(x) dx$$

$$= 2u_0 \int_{\mathcal{X}} [\int_{s} r_0(x)[\ddot{r}_1(x,s) - r_1(x)] p_0(s \mid x) ds] p_0(x) dx$$

$$= 2u_0 \int_{\mathcal{X}} [\int_{\mathcal{S}} r_0(x)[f_1^*(x) - q_1(x,\psi_\eta(s)) p_0(s \mid x) ds] p_0(x) dx \tag{21}$$

$$= 2u_0 \sqrt{\left| \int_{\mathcal{X}} [\int_{\mathcal{S}} r_0(x)[f_1^*(x) - q_1(x,\psi_\eta(s)) p_0(s \mid x) ds] p_0(x) dx \right|^2}$$

$$\leq 2u_0 \sqrt{\int_{\mathcal{X}} r_0(x)^2 p_0(x) dx \int_{\mathcal{X}} \int_{\mathcal{S}} [f_1^*(x) - q_1(x,\psi_\eta(s))]^2 p_0(s \mid x) ds p_0(x) dx}$$

$$= 2u_0 \sqrt{\epsilon_F^0 Q_0(\psi_\eta, q)}$$

$$H_4 \leq 2u_1 \sqrt{\epsilon_F^1 Q_1(\psi_\eta, q)} \tag{22}$$

Therefore, the bound of ITE risk is:

$$\epsilon_{ITE} \leq \epsilon_{PIP} + \mathbb{E}_{x \sim p(X)} [B \cdot IPM_G(p_0(\phi \mid x), p_1(\phi \mid x))] + (2\delta + 1)\tilde{\epsilon}_{CF} + 2 \sum_{t=0,1} u_t \left[ \sqrt{\epsilon_F^t Q_t(\psi_\eta, q)} \right]. \tag{23}$$

$\square$

## A.3 PROOF OF PROPOSITION A.3

**Proposition** (Relation between IPM and KL Divergence). *Let $G$ be the family of norm-1 functions in a Reproducing Kernel Hilbert Space (RKHS). Assume $G$ is generated by a normalized kernel. The*

*Integral Probability Metric (IPM) between two distributions $p_0(\phi \mid x)$ and $p_1(\phi \mid x)$) is bounded by their conditional KL divergence as follows:*

$$IPM_G(p_0(\phi \mid x), p_1(\phi \mid x)) \leq \sqrt{\frac{2}{\pi_0 \pi_1} \mathbb{E}_{\phi \sim p(\phi|x)} \big[ KL(p(t \mid x, \phi) \| p(t \mid x)) \big]},$$

*where $\pi_0 = p(t = 0 \mid x), \pi_1 = p(t = 1 \mid x)$. Specifically, if $\mathbb{E}_{s,x \sim p(s,x)}[KL(p(t \mid x, \phi) \| p(t \mid x))] = 0$, then $\mathbb{E}_{x \sim p(x)}[IPM_G(p_0(\phi \mid x), p_1(\phi \mid x))] = 0$.*

*Proof.* In the following, we denote $\pi_0 = p(t = 0 \mid x), \pi_1 = p(t = 1 \mid x), M = \pi_1 p_1 + \pi_0 p_0, p_1 = p_1(\phi \mid x), p_0 = p_0(\phi \mid x)$. And $JS_\pi(p_1 \| p_0)$ denote the Jensen-Shannon Divergence as follows:

$$JS_\pi(p_1 \| p_0) = \sum_{i=0}^{1} \pi_i KL(p_i \| M).$$

Our proof includes three steps:

1. prove $\mathbb{E}_\phi[KL[p(t \mid x, \phi) \| p(t \mid x)]] = JS_\pi(p_1 \| p_0)$;

2. prove $IPM_G(p_0(\phi \mid x), p_1(\phi \mid x)) \leq \sqrt{8\mathbb{E}_{\phi \sim p(\phi|x)} \big[ KL(p(t \mid x, \phi) \| p(t \mid x)) \big]}$;

3. prove $\mathbb{E}_{s,x \sim p(S,X)}[KL(p(t \mid x, \phi) \| p(t \mid x))] = 0 \Rightarrow \mathbb{E}_{x \sim p(X)}[IPM_G(p_0(\phi \mid x), p_1(\phi \mid x))] = 0$.

The detailed steps are as follows:

**Step 1: Prove $\mathbb{E}_\phi[KL[p(t \mid x, \phi) \| p(t \mid x)]] = JS(p_1 \| p_0)$:**

With the following equations,

$$\begin{aligned}
&\mathbb{E}_{\phi \sim p(\phi|x)}\big[\text{KL}(p(t|x, \phi) \| p(t|x))\big] \\
&= I(t; \phi|x) \\
&= H(\phi|x) - H(\phi|x, t) \\
&= H(p(\phi|x)) - \mathbb{E}_{t \sim p(t|x)}[H(\phi|x, t)] \\
&= H(M) - \big[\pi_1 H(p_1) + \pi_0 H(p_0)\big]
\end{aligned}$$

$$\begin{aligned}
&JS(p_1 \| p_0) \\
&= \pi_1 \text{KL}(p_1 \| M) + \pi_0 \text{KL}(p_0 \| M) \\
&= \pi_1(-H(p_1) - \mathbb{E}_{p_1}[\log M]) + \pi_0(-H(p_0) - \mathbb{E}_{p_0}[\log M]) \\
&= -(\pi_1 H(p_1) + \pi_0 H(p_0)) - (\pi_1 \mathbb{E}_{p_1}[\log M] + \pi_0 \mathbb{E}_{p_0}[\log M]) \\
&= -(\pi_1 H(p_1) + \pi_0 H(p_0)) - \mathbb{E}_M[\log M] \\
&= H(M) - \big[\pi_1 H(p_1) + \pi_0 H(p_0)\big]
\end{aligned}$$

Then we have $\mathbb{E}_{\phi \sim p(\phi|x)}\big[\text{KL}(p(t|x, \phi) \| p(t|x))\big] = JS_\pi\big(p(\phi|x, t = 1) \| p(\phi|x, t = 0)\big)$.

**Step 2: Prove $IPM_G(p_0(\phi \mid x), p_1(\phi \mid x)) \leq \sqrt{\frac{2}{\pi_0 \pi_1}\mathbb{E}_{\phi \sim p(\phi|x)}\big[KL(p(t \mid x, \phi) \| p(t \mid x))\big]}$:**

Combining Lemma 5 and Theorem 7 from (Hoyos-Osorio & Sanchez-Giraldo, 2023) yields the following inequality for any kernel $\kappa$ satisfying $\kappa(x, x) = 1$:

$$JS(P \| Q) \geq \frac{\pi_0 \pi_1}{2} IPM_{\kappa^2}^2(P, Q).$$

Since $G$ is generated by a normalized kernel, then we have:

$$JS(p_1\|p_0) \geq \frac{\pi_0 \pi_1}{2} IPM_G^2(p_0, p_1).$$

Therefore, we have $IPM_G(p_0, p_1) \leq \sqrt{\frac{2}{\pi_0 \pi_1} \mathbb{E}_{\phi \sim p(\phi|x)} \big[ KL(p(t \mid x, \phi) \| p(t \mid x)) \big]}$

**Step 3: Prove** $\mathbb{E}_{s,x \sim p(s,x)}[KL(p(t \mid x, \phi) \| p(t \mid x))] = 0 \Rightarrow \mathbb{E}_{x \sim p(x)}[IPM_G(p_0(\phi \mid x), p_1(\phi \mid x))] = 0$**:**

Since the KL divergence is non-negative, the expectation $E_{\phi,x \sim p(\phi,x)}[KL(p(t \mid x, \phi) \| p(t \mid x))] = 0$ implies that $KL(p(t \mid x, \phi) \| p(t \mid x)) = 0$ for all $\phi, x$ in the support of $p(\phi, X)$ (or almost everywhere with respect to $p(\Phi, X)$).

Furthermore, for any given $x$ and $\phi$, $KL(p(t \mid x, \phi) \| p(t \mid x)) = 0$ if and only if $p(t \mid x) = p(t \mid x, \phi)$ for all $t \in \{0, 1\}$. This equality, $p(t \mid x) = p(t \mid x, \phi)$, directly leads to conditional independence. By the definition of conditional probability, $p(t, \phi \mid x) = p(t \mid \phi, x)p(\phi \mid x)$. Since $p(t \mid \phi, x) = p(t \mid x)$, we have $p(t, \phi \mid x) = p(t \mid x)p(\phi \mid x)$. This factorization is the definition of conditional independence, $t \perp\!\!\!\perp \phi \mid x$. The conditional independence $t \perp\!\!\!\perp \phi \mid x$ implies that the distribution of $\phi$ given $x$ is independent of the value of $t$. Specifically, $p(\phi \mid x, t) = p(\phi \mid x)$ for all $t$. For $t \in \{0, 1\}$, this means $p(\phi \mid x, t = 0) = p(\phi \mid x, t = 1)$ almost everywhere with respect to the measure for $\phi$. Thus, $p_0(\phi \mid x) = p_1(\phi \mid x)$ almost everywhere for each $x$.

From the definition of the Integral Probability Metric (IPM), we know that $IPM_G(P_1, P_2) = 0$ if and only if $P_1 = P_2$ (for a suitable class of functions $G$). Therefore, for every $x$, since the conditional distributions $p_0(\phi \mid x)$ and $p_1(\phi \mid x)$ are equal as distributions over $\phi$, we have $IPM_G(p_0(\phi \mid x), p_1(\phi \mid x)) = 0$.

Finally, taking the expectation over $x$, we obtain $\mathbb{E}_{x \sim p(x)}[IPM_G(p_0(\phi \mid x), p_1(\phi \mid x))] = 0$. $\square$

### A.4 PROOF OF EXAMPLE 1

*Proof.* First, we define the true counterfactual outcome is

$$y(1 - t) = X + \alpha_2(1 - T) + S(1 - T) + \mathcal{N}_{cf}(0, 1), \tag{24}$$

where $S(1 - T) = X + \alpha_1(1 - T) + u_s$. $\mathcal{N}_{cf}(0, 1)$ is an independent noise term in $y(1 - t)$. so

$$\begin{aligned} y(1 - t) &= X + \alpha_2(1 - T) + (X + \alpha_1(1 - T) + u_s) + \mathcal{N}_{cf}(0, 1) \\ &= 2X + (\alpha_1 + \alpha_2)(1 - T) + u_s + \mathcal{N}_{cf}(0, 1). \end{aligned} \tag{25}$$

1. **Regress from** $(X, T)$**:** The predictor is $\hat{y}(1 - t) = \mathbb{E}[y(1 - t)|X, T]$. Given $y(1 - t) = 2X + (\alpha_1 + \alpha_2)(1 - T) + u_s + \mathcal{N}_{cf}(0, 1)$, and since $\mathbb{E}[u_s|X, T] = 0$ and $\mathbb{E}[\mathcal{N}_{cf}(0, 1)|X, T] = 0$ due to independence,
$$\hat{y}(1 - t) = 2X + (\alpha_1 + \alpha_2)(1 - T).$$

The prediction error is:

$$\begin{aligned} \hat{y}(1 - t) - y(1 - t) =&(2X + (\alpha_1 + \alpha_2)(1 - T)) \\ &- (2X + (\alpha_1 + \alpha_2)(1 - T) + u_s + \mathcal{N}_{cf}(0, 1)) \\ =&- u_s - \mathcal{N}_{cf}(0, 1). \end{aligned}$$

The error mean is $\mathbb{E}[-u_s - \mathcal{N}_{cf}(0, 1)] = 0$. The error variance is $Var(-u_s - \mathcal{N}_{cf}(0, 1)) = Var(u_s) + Var(\mathcal{N}_{cf}(0, 1)) = \sigma_u^2 + 1$. The prediction error $\hat{y}(1 - t) - y(1 - t)$ follows a $\mathcal{N}(0, \sigma_u^2 + 1)$ distribution.

2. **Regress from** $(X, T, S)$**:** Given observed $S = X + \alpha_1 T + u_s$. A model-based predictor $\hat{y}(1 - t)$ is formed from the structure of the model for $Y = X + \alpha_2 T + S + \mathcal{N}(0, 1)$ by substituting $1 - T$ for $T$ and using the observed $S$:

$$\begin{aligned} \hat{y}(1 - t) &= X + \alpha_2(1 - T) + S \\ &= X + \alpha_2(1 - T) + (X + \alpha_1 T + u_s) \\ &= 2X + \alpha_2(1 - T) + \alpha_1 T + u_s. \end{aligned}$$

The prediction error, using the true counterfactual outcome $y(1-t) = 2X + (\alpha_1 + \alpha_2)(1-T) + u_s + \mathcal{N}_{cf}(0,1)$, is:

$$
\begin{aligned}
\hat{y}(1-t) - y(1-t) =& (2X + \alpha_2(1-T) + \alpha_1 T + u_s) \\
& - (2X + (\alpha_1 + \alpha_2)(1-T) + u_s + \mathcal{N}_{cf}(0,1)) \\
=& \alpha_2(1-T) + \alpha_1 T - (\alpha_1 + \alpha_2)(1-T) - \mathcal{N}_{cf}(0,1) \\
=& \alpha_1(2T-1) - \mathcal{N}_{cf}(0,1).
\end{aligned}
$$

The mean of this error is $\mathbb{E}[\alpha_1(2T-1) - \mathcal{N}_{cf}(0,1)] = \alpha_1(2T-1)$ (assuming $E[\mathcal{N}_{cf}] = 0$ and $2T-1$ is treated as constant or $\alpha_1 = 0$). The error centered by its mean is $(\hat{y}(1-t) - y(1-t)) - \mathbb{E}[\hat{y}(1-t) - y(1-t)] = -\mathcal{N}_{cf}(0,1)$. The variance of the error centered by its mean is $Var(-\mathcal{N}_{cf}(0,1)) = 1$. The prediction error $\hat{y}(1-t) - y(1-t)$ follows a $\mathcal{N}(\alpha_1(2T-1), 1)$ distribution.

3. **Regress from** $(X, T, u_s)$**:** The predictor is $\hat{y}(1-t) = \mathbb{E}[y(1-t)|X, T, u_s]$. Given $y(1-t) = 2X + (\alpha_1 + \alpha_2)(1-T) + u_s + \mathcal{N}_{cf}(0,1)$. Since $E[\mathcal{N}_{cf}(0,1)|X, T, u_s] = 0$ as $\mathcal{N}_{cf}(0,1)$ is independent of $X, T, u_s$,

$$\hat{y}(1-t) = 2X + (\alpha_1 + \alpha_2)(1-T) + u_s. \tag{26}$$

The prediction error is:

$$
\begin{aligned}
\hat{y}(1-t) - y(1-t) =& (2X + (\alpha_1 + \alpha_2)(1-T) + u_s) \\
& - (2X + (\alpha_1 + \alpha_2)(1-T) + u_s + \mathcal{N}_{cf}(0,1)) \\
=& -\mathcal{N}_{cf}(0,1).
\end{aligned}
$$

The mean of this error is $\mathbb{E}[-\mathcal{N}_{cf}(0,1)] = 0$. The variance of the error is $Var(-\mathcal{N}_{cf}(0,1)) = Var(\mathcal{N}_{cf}(0,1)) = 1$. The prediction error $\hat{y}(1-t) - y(1-t)$ follows a $\mathcal{N}(0,1)$ distribution.

$\square$

# B  DATASETS

## B.1  IHDP

The IHDP dataset is collected from randomized controlled trial (RCT) experiments designed to evaluate the effect of specialist home visits on the future cognitive test scores of premature infants. Hill (2011) removes a non-random subset of the treated group to induce selection bias. In total, IHDP comprises 747 units (139 treated, 608 control), each represented by 25 pre-treatment variables related to the children and their mothers. Please refer to (Hill, 2011) for more details.

Following (Cheng et al., 2021), we define the post-treatment variables for each unit $i$ as a time series $\{s_{ik}^{t_i}\}_{k=1}^K$ of length $K$. We consider $m$-dimensional post-treatment variables, $s_{ik}^{t_i} \in \mathbb{R}^m$, for each time step $k \in \{1, 2, \ldots, K\}$. These variables are generated based on the unit's treatment indicator $t_i \in \{0, 1\}$, baseline covariates $x_i \in \mathbb{R}^{1 \times 25}$, and previous values in the time series.

For $k = 1$, the initial post-treatment variable $s_{i1}^{t_i} \in \mathbb{R}^m$ for unit $i$ is directly taken from the observed outcome in the IHDP dataset (distinct from the variable $y$ in our model).

For time steps $k > 1$, the variable $s_{ik}^{t_i}$ is generated based on the unit's treatment indicator $t_i \in \{0, 1\}$, baseline covariates $x_i \in \mathbb{R}^{25}$, and previous values in the time series $\{s_{ij}^{t_i}\}_{j=1}^{k-1}$. The generative model for $k > 1$ is as follows:

$$
s_{ik}^{t_i} = \begin{cases} x_i\beta_0 + \frac{C_1}{k-1}\sum_{j=1}^{k-1} s_{ij}^{t_i} + \epsilon_u, & t_i = 0 \\ x_i\beta_1 + \frac{C_1}{k-1}\sum_{j=1}^{k-1} s_{ij}^{t_i} + \epsilon_u, & t_i = 1 \end{cases} \quad \text{for } k > 1. \tag{27}
$$

Here, $\beta_{t_i}$ represents the coefficient matrix specific to the treatment group ($\beta_0 \in \mathbb{R}^{25 \times m}$ if $t_i = 0$ and $\beta_1 \in \mathbb{R}^{25 \times m}$ if $t_i = 1$) and $C_1$ is a scalar scaling factor. The exogenous noise vectors $\epsilon_u \in \mathbb{R}^m$ are generated for $k > 1$ by sampling each of their $m$ elements independently from the Laplace distribution with mean 0 and scale 1.

The coefficient matrices $\beta_0 \in \mathbb{R}^{25 \times m}$ and $\beta_1 \in \mathbb{R}^{25 \times m}$ are generated by sampling each of their elements independently according to the specified discrete probability distributions:

- Elements of $\beta_0$: sampled from $\{0, 1, 2, 3, 4\}$ with probabilities $\{0.5, 0.2, 0.15, 0.1, 0.05\}$, respectively.

- Elements of $\beta_1$: sampled from $\{-2, -1, 0, 1, 2\}$ with probabilities $\{0.2, 0.2, 0.2, 0.2, 0.2\}$, respectively.

The outcome is defined as $Y_{t_i}(x_i) = \frac{1}{3N} \sum_{k=K-2}^{K} \sum_j^N s_{jk}^{t_i}$. We randomly split the samples into train/validation/test with a 60/20/20 ratio.

## B.2 NEWS

The News dataset simulates a media consumer's opinions on various news items, viewed either on a mobile device or desktop. Each news item is represented by word counts $x_i \in \mathbb{N}^V$, where $V$ is the number of words. The outcome $y_i \in \mathcal{R}$ reflects reader experience, and the treatment $t_i \in \{0, 1\}$ indicates device: desktop ($t = 0$) or mobile ($t = 1$). A topic model trained on a large document set represents consumer preferences. Let $k$ be the number of topics and $z(x) \in \mathcal{R}^k$ be the topic distribution of news item $x$, with $z_1^c$ (mobile) and $z_0^c$ (desktop) as centroids in the topic space. The outcome is determined by the similarity between $z(x_i)$ and $z_t^c$: $y_i^{t_i} = C(z(x_i)^T z_0^c + t_i \cdot z(x_i)^T z_1^c) + \epsilon$, where $C$ is a scaling factor and $\epsilon \sim N(0, 1)$ is noise. In total, News comprises 5,000 news items, each with 3,477 word counts. For more details, please refer to (Johansson et al., 2016).

We define the post-treatment variables for each unit $i$ as a time series $\{s_{ik}^{t_i}\}_{k=1}^K$ of length $K$. We consider multidimensional $s_{ik}^{t_i} \in \mathbb{R}^{m \times 1}$ as follows:

$$s_{ik}^{t_i} = C_2 \left( z(x_i)^\top z_0^c + t_i \cdot z(x_i)^\top z_1^c \right) + A s_{i(k-1)}^{t_i} + \epsilon_u, \tag{28}$$

where $\epsilon_u$ represents exogenous noise generated from the Laplace distribution with mean 0 and scale 1. $C_2$ is a scaling factor and $A \in \mathbb{R}^{m \times m}$ is the transfer matrix with eigenvalue less than 0.9. For IHDP, the outcome is defined as the last post-treatment variable. The outcome is defined as $Y_{t_i}(x_i) = \frac{1}{3N} \sum_{k=K-2}^{K} \sum_j^N s_{jk}^{t_i}$. Table 1 presents the results for $K = 60$. We randomly split the samples into train/validation/test with a 60/20/20 ratio.

## B.3 SYNTHETIC

The synthetic dataset simulates a temporal causal system with interacting variables over time. The initial state variables $x_{i0} \in \mathbb{R}^N$ are sampled from a standard normal distribution. The state evolution follows:

$$x_{i(k+1)} = x_{ik} + a_x t_{ik} + \epsilon_x, \tag{29}$$

where $a_x \in \mathbb{R}^{N \times 1}$ is a coefficient matrix. The binary treatment $t_{ik}$ is assigned through:

$$t_{ik} \sim \text{Bernoulli}(\text{sigmoid}(x_{ik}\beta_t + \epsilon_t)), \tag{30}$$

where $\beta_t \in \mathbb{R}^N$ are treatment coefficients. At each timestep $k$, three types of post-treatment variables are generated:

$$v_{ik} = \beta_v x_{ik} + \gamma_v t_{ik} + \epsilon_v, \quad v_{ik} \in \mathbb{R}^N \tag{31}$$

$$m_{ik} = \beta_m x_{ik} + \gamma_m t_{ik} + \epsilon_m, \quad m_{ik} \in \mathbb{R}^N \tag{32}$$

$$a_{ik} = \beta_a x_{ik} + \epsilon_a, \quad a_{ik} \in \mathbb{R}^N \tag{33}$$

where $\beta_v, \beta_m, \beta_a \in \mathbb{R}^{N \times N}, \gamma_v, \gamma_m \in \mathbb{R}^{1 \times N}$ are coefficient matrices. The post-treatment variables are concatenated as $s_{ik} = [v_{ik}; m_{ik}; a_{ik}] \in \mathbb{R}^{3N}$. The final outcome is determined by:

$$y_i = \frac{1}{C} \left( \sum_{k=k_0}^K \gamma_y^{K-k}(x_{ik}\beta_y + m_{ik}\beta_m + a_{ik}\beta_a)(1 + \epsilon_u) + t_{iK}\beta_t \right), \tag{34}$$

where $\epsilon_u$ represents exogenous noise sampled from Laplace distribution, $\gamma_y = 0.99$ is a decay factor, $k_0 = K - \lfloor \alpha K \rfloor$ with $\alpha$ controlling the temporal window, $\beta_y, \beta_m, \beta_a \in \mathbb{R}^{N \times 1}$ are outcome coefficients for state, mediator and adjustment variables respectively, $\beta_t \in \mathbb{R}$ is the direct treatment

effect coefficient, and $C$ is a normalizing constant. All variables are normalized using StandardScaler. All noise terms $\epsilon$ are sampled from Laplace distributions.

We simulated 10,000 samples for this dataset. To validate the Impact of Task Complexity on PIPCFR, we conduct experiments with different values of $K$ and $\epsilon$ as shown in Section 5.2. Table 1 presents the results for $K = 60$. We randomly split the samples into train/validation/test with a 60/20/20 ratio.

## C  TRAINING DETAILS

We implement PIPCFR using PyTorch, employing a TARNET architecture for the student network and CFRNet (Shalit et al., 2017) for the teacher network. During training, we use Adam (Kingma, 2014) optimizer with a learning rate of 0.001 and a decay rate of 0.95. The batch size is set to 250. Our network architecture consists of 3 Multi-Layer Perceptron (MLP) layers for covariate representations and 4 layers for hypothesis functions, with all layers maintaining a consistent size of 64 units across all datasets. For the post-treatment variable representation function $\psi_\eta$, we utilize 3 MLP layers with a hidden dimension of 128. The propensity score models ($g$ and $\tilde{g}$) comprise 4 MLP layers followed by a sigmoid activation function. We set the KL loss weight $\gamma$ to 1 for all datasets, computing all losses per mini-batch at each iteration. Following CFR (Shalit et al., 2017), we implement two versions of our algorithm using different IPM distance metrics: PIPCFR(MMD) based on Maximum Mean Discrepancy (Gretton et al., 2012) and PIPCFR(WASS) using Wasserstein distance (Cuturi & Doucet, 2014). Please refer to Shalit et al. (2017) for the computation of these distances. All methods were trained using a V100 GPU with 32GB of VRAM.

