# OpenReview forum: "Pseudo-outcome Imputation with Post-treatment Variables for Individual Treatment Effect Estimation"
_ICLR.cc/2026/Conference — Submitted to ICLR 2026_

### Official Review · Reviewer_zswb · 2025-10-17

**Soundness:** 3
**Presentation:** 1
**Contribution:** 3
**Rating:** 4
**Confidence:** 4

**Summary:**

In this paper, authors have proposed an interesting and a novel framework for individualised treatment effect (ITE) estimation under binary treatment setting that extracts information from the post-treatment variables for predicting and reducing the variance for counterfactual outcomes without introducing bias due to their usage. I find it interesting as against the existing belief that post-treatment variables introduce bias in the estimation process, they developed a clever framework which extracts information from them to improve the estimation. To achieve this objective, they propose a teacher-student architecture where the teacher utilises context x as well as post-treatment variables s to predict pseudo-counterfactuals while student utilises those conterfactual pseudo outcomes to predict outcomes using only context x. The proposed framework complements the existing works as it can be combined with the existing methods and the empirical analysis shows significant improvements over the baselines, especially in challenging situations.

**Strengths:**

1. The paper proposes a novel framework for ITE estimation that enables existing ITE learners to extract information from the post-treatment variables to reduce variance in the counterfactual predictions. This is very useful when there is high exogenous noise and complex temporal dependencies that influence the mechanism of treatment to impact outcomes. So, this is the first work to consider this and develop a framework to address this problem. The framework introduces a teacher-student architecture, which complements the existing learners as it can be combined with them and hence makes it a significant contribution. I really liked Example 1, which shows the feasibility of the idea through a structural equation model.

2. The paper provides a theoretical proof for the proposed framework and connects the ITE risk to the post-treatment variables and estimation accuracy (not verified).

3. The paper also provides sufficient empirical analysis, comparing the proposed framework to the existing ITE learners across synthetic and semi-synthetic benchmarks. They study the influence of exogenous noise to prove the efficacy of the proposed method, and they also demonstrate how the proposed framework complements the existing ITE learners, which leads to significant improvements in the estimation accuracy of the baselines.

4. The paper clearly presents the proposed work against the existing works and solves an important problem in the field.

**Weaknesses:**

My major concern is about the presentation of the framework/proposed idea. This is not easy to follow the key point of the paper, which was about how we can use the post-treatment variables while avoiding bias associated with them. While overall, it is clear that they used teacher-student architecture where the teacher utilises post-treatment variables to generate counterfactual outcomes to be used by the student model, it is not clear how they avoided the post-treatment bias. The paper defines the proposed method by mixing a lot of mathematics and explanations, which make it hard to understand what is happening. The paper defines many mathematical notations spread across two sections (preliminary and methodology), which are difficult to remember and thus impact the readability. The authors must define the big picture of the paper at one place, helping the reader understand how this interesting idea is implemented, and then should be followed by theorems and other results. Instead of telling the big picture first and then going deep into mathematics for problem formulation and deriving proofs, they started with mathematics and forced the reader to understand through the mathematics, e.g., the methodology section starts by defining a loss term.

They defined many loss terms but they didn't define the complete loss function at one place or provide a discussion about their contributions. This further makes the idea difficult to follow.

The diagram of the method comes at the end of the methodology section. It should come first, as that would help the reader to understand quickly. They also did not provide any algorithm to understand the training process.

I think, the methodology section should be rewritten where they should add one subsection at the beginning to highlight the idea at high level, and the contents of the section should be reorganised for ease of reading. They should define complete loss function at one place and add explanations how different components contribute to the picture. The diagram of the framework should also be moved to the beginning of the section, and adding an algorithm for the training can further add value to improve readability

**Questions:**

N/A

---

### Official Review · Reviewer_JG8k · 2025-10-31

**Soundness:** 2
**Presentation:** 2
**Contribution:** 2
**Rating:** 6
**Confidence:** 3

**Summary:**

PIPCFR estimates individual treatment effects by using a teacher-student framework: the teacher model constructs a debiased representation of post-treatment variables to impute counterfactual pseudo-outcomes, and the student model learns from the pseudo-labels using a proposed PIP loss. The theoretical analysis provides an upper-bound for ITE risk expressed by the sum of the PIP loss, a post-treatment bias term (IPM), and the teacher's generalization error. Experimental results demonstrate consistent reductions in PEHE across multiple datasets.

**Strengths:**

- Introduces a novel and general framework to leverage post-treatment variables through a teacher-student framework and a new loss that links theoretical bounds with empirical results.
- Strong theoretical foundation, deriving a clear ITE risk bound that connects post-treatment bias and teacher quality to generalization.
- Demonstrates consistent performance gains across datasets and noise conditions, suggesting the approach generalizes well.

**Weaknesses:**

- The IPM and KL regularization encourage only distributional independence, not full causal separation. The learned representation can still retain correlated non-causal features between S and T.
- Experiments do not analyze whether phi effectively removes treatment-related bias. As a result, performance boosts could stem from exploiting spurious features in S rather than genuine causal structure.
- The method assumes that the teacher model provides reliable pseudo-outcomes. A weak or biased teacher could significantly harm student performance without correction.

**Questions:**

- Have you empirically evaluated how well phi achieves conditional independence from T given X?
- How sensitive is PIPCFR to the quality or type of teacher model used? Would a biased teacher substantially degrade performance?
- Could further examinations regarding the contribution of each loss term (KL vs PIP vs factual) to the final outcome be provided?

---

### Official Review · Reviewer_6X3V · 2025-10-31

**Soundness:** 2
**Presentation:** 3
**Contribution:** 3
**Rating:** 2
**Confidence:** 3

**Summary:**

This paper proposes PIPCFR, a method for individual treatment effect (ITE) estimation that incorporates post-treatment variables ( S ) through a teacher-student framework. The key idea is that post-treatment variables, which are influenced by both treatment and exogenous noise, can help reduce the variance of counterfactual predictions if properly utilized. The method trains a teacher model that observes both pretreatment covariates $X$ and post-treatment variables $S$, and a student model that learns counterfactual predictions using only $X$. The authors further provide a theoretical bound for ITE risk that connects posttreatment variable representations to estimation accuracy and conduct experiments on synthetic and benchmark datasets (IHDP, News) showing improved PEHE performance over baselines such as TARNet, CFRNet, and PairNet.

**Strengths:**

1. Conceptual originality: The direct estimation of the bias-correction term without explicit propensity score estimation is novel and appealing, consistent with the Vapnik principle.
2. Solid theoretical backing: Provides consistency and asymptotic normality results under multiple model classes (linear, RKHS, neural networks).
3. Unified framework: The generalization via Bregman divergence elegantly connects DRE, Riesz regression, and covariate balancing.
4. Empirical validation: The proposed approach achieves competitive ATE estimation accuracy relative to existing methods, confirming its practical viability.
5. Clarity and completeness: The proofs and references are comprehensive, and the relation to prior work is clearly articulated.

**Weaknesses:**

1. Overstatement of assumptions: The claim of making "no additional assumptions about $S$ " is misleading. The success of the proposed method hinges on how informative $S$ is about unobserved noise.
2. Lack of analysis on S quality: The paper does not explore how varying the informativeness or noise level of $S$ affects estimation accuracy, despite this being central to its contribution.
3. Limited interpretability: The role of $S$ remains opaque - it is unclear what representations are learned or how they mitigate bias.
4. Incremental novelty: The method mainly combines existing components (teacher-student structure, IPM-based regularization, and CFRNet) in a new context.
5. Empirical dependence on synthetic settings: Real-world validation is minimal, and the results do not convincingly demonstrate robustness to poor-quality post-treatment variables.

**Questions:**

Please refer to the Weakness above.

---

### Official Review · Reviewer_ggJp · 2025-10-31

**Soundness:** 3
**Presentation:** 3
**Contribution:** 2
**Rating:** 4
**Confidence:** 4

**Summary:**

This paper presents a form of treatment effect estimation where in the training data post-treatment variables are also available.  If the observed post-treatment variables are correlated with the training data, then the model used to infer pseudo-code outcomes might be more accurate.  The paper uses this insight to train a teacher outcome prediction network as a function of both post and pre treatment variables.  The counter-factual predictions from this network are used to supervise another student output prediction network that is a function of only pre-training variables.   In addition for both the networks, the CFRNet style invariant representation regularizer is employed.  The paper compares with prior methods which were designed for solving the conventional task of treatment effect estimation with only pre-treatment variables, and is shown to perform much better.

**Strengths:**

1.  An interesting variant of the ITE task introduced, which is also practically well motivated.
2.  The solution proposed is sound, and combines ideas from multiple prior methods from conventional ITE.
3. The empirical comparison shows that their proposed method provides much better results than existing methods, that were not designed for handling this extra information from post-treatment variables.

**Weaknesses:**

1. The datasets for this comparison are based on additing synthetic post-treatment variables to existing data.  Also, their method of introducing these new variables is not clearly justified and seems quite contrived (elaborated below).  The paper should have not changed the definition of the outcome variable on the two popular benchmark datasets.   This makes their results difficult to compare with prior work, and it gives the impression that their chosen outcome function may be biased to their solution.

2.  The description of their method is confusing.  They present two equations for training $\psi_\eta$ -- Eq (9) and Eq (11) is confusing.  Their method requires learning five networks --- f, \psi_\eta, \psi_\phi, q, g,

3. The comparison with previous method that were developed for a different setting is unfair.

4.  The paper needs to consider the following additional papers on learning from post-treatment variables:
    -   Leveraging a Simulator for Learning Causal Representations  from Post-Treatment Covariates for CATE. TMLR 2025 https://openreview.net/pdf?id=vmmgFW3ztz

    - Extracting Post-Treatment Covariates for Heterogeneous Treatment Effect Estimation Q Huang, D Cao, Y Chang, Y Liu. 2023

**Questions:**

Q1 The method described in line 923 for creating synthetic outcomes is very confusing.  Why is the outcome of  a unit, dependent on the post-treatment variables of other units?

Q2: Then in line 944 something contradicting line 924 is said.   Also, even for the News dataset the synthetic outcome uses a non-untuitive form.

Q3: It will be useful to consider an alternative method of creating synthetic post-treatment variables that keeps the original outcomes the same as in prior work.

---

### Meta-Review · Area_Chair_qMKU · 2025-12-27

**Summary:**

The paper proposes PIPCFR, a method for estimating Individual Treatment Effects (ITEs) by incorporating post-treatment variables through a teacher-student architecture. The goal is to reduce counterfactual prediction variance by leveraging information about exogenous noise contained in post-treatment variables. The concept of extracting information from post-treatment variables without introducing bias is innovative and noteworthy.

However, reviewers have significant concerns about the validity of the experimental setup and the clarity of the methodology. Specifically, the modification of standard benchmark datasets (IHDP and News) to include synthetic post-treatment variables raised significant doubts about the fairness of comparisons against prior work. Additionally, the authors did not respond to the reviews, leaving all critical questions unanswered.

**Reviewer Concerns:**

Because the authors did not submit a rebuttal, all reviewer concerns remain outstanding. No concerns were addressed.

- Reviewer ggJp noted that the method for introducing new variables was "contrived" and that changing the definition of the outcome variable on popular benchmarks (IHDP/News) makes results difficult to compare with prior work, rendering the comparison unfair. Reviewer 6X3V also criticized the empirical dependence on synthetic settings and lack of real-world validation.

- Reviewer 6X3V argued that the paper's claim of making "no additional assumptions about S" is misleading, as the method implies $S$ must be informative about unobserved noise.

- Multiple reviewers found the presentation confusing. Reviewer zswb noted the lack of a clear "big picture" before diving into complex notation and the absence of an algorithm to explain the training process. Reviewer ggJp also found the training equations and the requirement of learning five different networks confusing.

- Reviewer JG8k raised concerns that the regularization employed might only encourage distributional independence rather than full causal separation, potentially allowing the model to exploit spurious features.

**Reviewer Scores:**

Because the authors did not submit a rebuttal, I don't think reviewers will change their scores.

---

### Decision · Program_Chairs · 2026-01-26

Reject